# Monitoring the Intracellular pH and Metabolic State of Cancer Cells in Response to Chemotherapy Using a Combination of Phosphorescence Lifetime Imaging Microscopy and Fluorescence Lifetime Imaging Microscopy

**DOI:** 10.3390/ijms25010049

**Published:** 2023-12-19

**Authors:** Irina Druzhkova, Anastasiya Komarova, Elena Nikonova, Vadim Baigildin, Artem Mozherov, Yuliya Shakirova, Uliana Lisitsa, Vladislav Shcheslavskiy, Nadezhda Ignatova, Evgeny Shirshin, Marina Shirmanova, Sergey Tunik

**Affiliations:** 1Institute of Experimental Oncology and Biomedical Technologies, Privolzhsky Research Medical University, 603005 Nizhny Novgorod, Russia; komarova.anastasii@gmail.com (A.K.); artemmozherov@gmail.com (A.M.); u.a.lisitsa@gmail.com (U.L.); vis@becker-hickl.de (V.S.); n.i.evteeva@gmail.com (N.I.); shirmanovam@gmail.com (M.S.); 2Institute of Biology and Biomedicine, Lobachevsky State University of Nizhny Novgorod, 603950 Nizhny Novgorod, Russia; 3Laboratory of Clinical Biophotonics, Sechenov First Moscow State Medical University (Sechenov University), 119991 Moscow, Russia; nikonova87@yandex.ru (E.N.); eshirshin@gmail.com (E.S.); 4Institute of Chemistry, Saint-Petersburg State University, 198504 St. Petersburg, Russia; v.baygildin@spbu.ru (V.B.); y.r.shakirova@spbu.ru (Y.S.); 5Faculty of Physics, Lomonosov Moscow State University, 119991 Moscow, Russia

**Keywords:** intracellular pH, metabolism, HCT116 cell line, collagen, FOLFOX, PLIM, FLIM

## Abstract

The extracellular matrix (ECM), in which collagen is the most abundant protein, impacts many aspects of tumor physiology, including cellular metabolism and intracellular pH (pHi), as well as the efficacy of chemotherapy. Meanwhile, the role of collagen in differential cell responses to treatment within heterogeneous tumor environments remains poorly investigated. In the present study, we simultaneously monitored the changes in pHi and metabolism in living colorectal cancer cells in vitro upon treatment with a chemotherapeutic combination, FOLFOX (5-fluorouracil, oxaliplatin and leucovorin). The pHi was followed using the new pH-sensitive probe BC-Ga-Ir, working in the mode of phosphorescence lifetime imaging (PLIM), and metabolism was assessed from the autofluorescence of the metabolic cofactor NAD(P)H using fluorescence lifetime imaging (FLIM) with a two-photon laser scanning microscope. To model the ECM, 3D collagen-based hydrogels were used, and comparisons with conventional monolayer cells were made. It was found that FOLFOX treatment caused an early temporal intracellular acidification (reduction in pHi), followed by a shift to more alkaline values, and changed cellular metabolism to a more oxidative state. The presence of unstructured collagen markedly reduced the cytotoxic effects of FOLFOX, and delayed and diminished the pHi and metabolic responses. These results support the observation that collagen is a factor in the heterogeneous response of cancer cells to chemotherapy and a powerful regulator of their metabolic behavior.

## 1. Introduction

The extracellular matrix (ECM), which represents the complex network of macromolecules, such as collagens, proteoglycans and glycoproteins, is an important non-cellular component of a tumor. It is well documented that the ECM participates in the regulation of many aspects of tumor biology, including proliferation, differentiation, immunity, survival and migration. The most prominent feature of tumor ECM structures is highly aligned collagen fibers, which increases tissue stiffness [1]. ECM stiffness can affect cellular metabolism, and primarily, cellular glucose uptake, glycolysis and glycogen synthesis [2,3,4]. Also, it is known that ECM stiffness influences the intracellular signaling (e.g., cell stemness and malignancy) of cancer cells through biomechanical forces, apart from biochemical cues. Recently, it was demonstrated that a stiff microenvironment leads to increased expression of heat shock proteins, including hsp70 and hsp90, in cancer cells, resulting in better cell surveillance, an up-regulation of stemness marker expression in vitro and poorer responses to the treatment with doxorubicin in vivo [5]. The stiffness of tumor cells themselves is inversely correlated with their migration and invasion through 3D basement membranes [6].

The abundant, highly cross-linked ECM interferes with the efficacy of chemotherapy. Several studies show a protective role of collagen against cytotoxic chemotherapy, either via the mechanical restriction of drug penetration to tumor cells or by promoting cells’ resistance to apoptosis [7,8,9]. The latter is often associated with increases in ERK1/2 signaling, regulated by growth factors and integrins [10]. 

The rearrangement of cellular metabolism and dysregulated intra- and extracellular pH are the well-known hallmarks of cancer that play a crucial role in tumor progression and therapeutic response. The glycolytic shift in metabolism observed even in the presence of normal oxygen concentrations (the Warburg effect) is typically associated with more aggressive tumor phenotypes and a poor prognosis [11,12]. The alkaline intracellular pH (pHi) drives the upregulation of glycolysis, and the acidic pH of the extracellular medium promotes invasion and resistance to therapies [13,14]. The maintenance of pHi in a narrow range is a necessary condition for cellular homeostasis and survival, since many, if not all, cellular proteins and enzymes are pH-dependent [15]. At the same time, metabolic status is considered flexible, which allows cancer cells to adapt to the changeable microenvironment [16,17].

Cytotoxic chemotherapy affects both pHi homeostasis and the metabolic profile of tumor cells. Fluctuations in pHi precede cell death upon treatment and, therefore, accompany cell responses to anticancer drugs, while the capability of cancer cells to control pHi correlates with their survival [18,19,20,21,22]. The metabolic changes induced by chemotherapy usually include a switch from glycolysis to oxidative phosphorylation, which is presumably associated with the response to DNA damage and correlates with inhibited cell proliferation [21,22,23,24]. Although the pHi and cellular metabolism are tightly interconnected, these links may be disrupted upon chemotherapy. The role of the ECM in the regulation of cancer cell responses to cytotoxic drugs and, especially, its metabolic aspects have been explored poorly so far.

This study was aimed at the simultaneous monitoring of metabolic changes and pHi fluctuations in cancer cells in a 3D model with collagen during chemotherapy. Colorectal cancer cells were treated with the chemotherapeutic regimen FOLFOX (a combination of 5-fluorouracil, oxaliplatin and leucovorin), the gold standard treatment for colon cancer in patients [25,26]. In order to monitor the changes in pHi and metabolism in living cells, the combination of two methods based on the two-photon excited laser scanning microscopy was used: phosphorescence lifetime imaging (PLIM) and fluorescence lifetime imaging (FLIM). An obvious advantage of the combined use of PLIM and FLIM is that it allows the simultaneous investigation of the two parameters (pHi and metabolism) in the same cells, while making other spectral channels in the microscope available for the potential imaging of different fluorophores.

The assessment of pHi with PLIM is based on the use of a novel pH-sensitive cell-permeable phosphorescent probe [27], containing an iridium phosphorescent complex, which displays lifetime responses to variations in media acidity in biologically relevant intervals. Since phosphorescence is associated with the spin-forbidden triplet-to-singlet transitions, the phosphorescent emitters display large Stokes shifts, higher bleaching stability and a much longer lifetime (normally from microseconds to milliseconds) compared to fluorophores [28,29]. Consequently, PLIM can be easily combined with FLIM, even if phosphorescence and fluorescence are spectrally overlapped, due to the resolution in the time domain, thus providing new possibilities for multiparametric imaging [30]. Typically, phosphorescent dyes are inherently sensitive to molecular oxygen that quenches their triplet state, and therefore, PLIM is used to measure oxygen concentrations in cells and tissues [31,32,33,34]. This also means that pH-sensitive phosphorescent probes usually display crosstalk to the variations in oxygen concentration that may distort measurements of media acidity. However, the sensitivity to oxygen of the probe used in this study is blocked by packing the chromophore in a polymeric matrix, preserving the sensitivity to pH at an acceptable level.

Monitoring metabolism with FLIM is based on the detection of fluorescence lifetimes of the redox cofactor, reduced nicotinamide adenine dinucleotide (phosphate) (NAD(P)H). NAD(P)H exists in cells in two forms, free and protein-bound, whose ratio depends on the redox activity and metabolic state of the cells. Free NAD(P)H possesses a short fluorescence lifetime of around 0.4 ns and is associated with glycolysis, whereas the bound form has a lifetime from 1.5 to 4 ns, depending on its binding enzyme, a large portion of which are mitochondrial respiratory complexes [35,36]. The ratio of the free and bound NAD(P)H fractions serves as an indicator of the shifts in energy metabolism, which are often observed in response to anticancer treatment [21,22,23,24].

Here, we applied a multiparametric approach, based on the measurement of the NAD(P)H status with FLIM and pHi monitoring with PLIM to identify the effects of chemotherapy on cancer cells growing in the extracellular matrix.

## 2. Results

### 2.1. Effects of Collagen on Cytotoxicity of FOLFOX

According to the SHG analysis, the collagen in the 3D model was present as fine, thin, short fibers, which indicates that it was almost unstructured (Appendix A).

To evaluate the cytotoxic effects of the FOLFOX regimen in the presence of collagen, viability assays were used. The colony-forming assay revealed that cells extracted from collagen after 24 h of drug exposure formed the colonies more actively than the cells exposed to the drugs in the absence of collagen. The number of colonies was ~13 times higher (34.0 ± 4.4 vs. 2.5 ± 0.7, *p* = 0.0003), and the optical density of the crystal violet was 3.3 times higher (0.595 ± 0.055 vs. 0.180 ± 0.021, *p* = 0.000001) for the cells in collagen (Figure 1A). Since only a fraction of live cells retain the capacity to produce colonies, this result indicates the lower toxic effects of FOLFOX on cells cultured in collagen. We used the colony-forming assay only as a complementary test to the live/dead assay to confirm the greater efficacy of FOLFOX in the absence of collagen compared with the model where collagen is present. The test was performed once over 24 h of drug exposure.

The live/dead cell assay showed that the content of dead cells significantly increased after 48 h of incubation with FOLFOX in the absence of collagen. However, in the case of culturing in collagen, the number of dead cells was statistically lower. In the absence of collagen, the proportion of dead cells relative to the control increased ~2.7-fold after 24 h of incubation (0.63% vs. 0.23%) and ~24.2-fold after 48 h of incubation with the drugs (31.5% vs. 1.3%) (Figure 1B). In the presence of collagen, the content of dead cells after treatment was ~2.25 times higher than in the untreated control after 24 h (0.27% vs. 0.12%) and ~6 times higher after 48 h (0.36% vs. 0.06%).

### 2.2. Intracellular pH after Treatment with FOLFOX Regimen

The incubation of HCT116 cells with the solution of the BC-Ga-Ir probe resulted in the internalization of the probe and a gradual increase in the phosphorescence signal inside the cells. At the incubation times of 1–72 h, the intensity of the signal was sufficient for time-resolved measurements.

Colocalization experiments clearly indicated that the BC-GA-Ir probe is localized mainly in Golgi apparatus and the endoplasmic reticulum (the Manders overlap coefficient, M1: 0.855) and partly in the lysosomes (M1: 0.594) (Figure 2), which means that pH is measured mainly in these compartments. In cells without the BC-Ga-Ir probe, a phosphorescence signal was not detected (Appendix A).

It is important that the phosphorescence lifetimes (τ_m_) of BC-Ga-Ir in the untreated cells remained stable from 1 h to 72 h of incubation, and amounted to ~1.3 μs irrespective of the presence of collagen, corresponding to pH 6.87 (Appendix A). According to the literature, this value is typical of the cis-Golgi (pH ~6.7) [37]. Therefore, the obtained pH values are consistent with the localization of the probe in the Golgi apparatus.

### 2.3. Phosphorescence Lifetime Imaging Microscopy (PLIM)

In the FOLFOX-treated cells without collagen, the phosphorescence lifetime of BC-Ga-Ir significantly decreased, starting from 1 h, and was 1.12 μs 5 h after the addition of chemotherapeutic drugs, which indicates a decrease in pH to 4.75 units. After 24 h, the τ_m_ of the probe did not statistically differ from the control. Forty-eight and 72 h after the treatment, there was an increase in the τ_m_ to 1.46 μs and then to 1.62 μs, indicating a shift in pH to more alkaline values, 8.75 and 10.63, respectively (Figure 3).

In the model with collagen, a decrease in the phosphorescence lifetime to 1.2 μs (pH 5.69) was recorded later, at 5 h and 24 h of incubation with the drugs. Then, at 48 h and 72 h, the values of the BC-Ga-Ir τ_m_ did not differ from the untreated control (Appendix A).

It should be noted that, in general, the population of cells was heterogeneous in terms of the pHi, especially after the treatment. The majority of untreated cells had phosphorescence lifetimes of about 1.3 μs, but there were those with lifetimes >1.5 μs, and the number of the latter increased with time. In the treated groups, the number of cells with long lifetimes was notably higher, and at long incubation times, they presented the majority in the cell population (Figure 3B). The cells with extremely long phosphorescence lifetimes were the dead cells, as it is seen from their morphology (round shape, detachment from the surface) and the abnormal, homogeneous distribution of the pH probe.

If we consider only the sub-population of viable cells, the phosphorescence lifetime first decreased to 1.2 μs (pH 5.69) in the period from 1 h to 5 h and then increased to 1.5 μs (pH 9.2) in the period from 48 h to 72 h of incubation with the drugs (Appendix A). Thus, the late increase in the phosphorescence lifetime as a result of treatment was due to both the contribution from viable cells and the increased proportion of dead cells. Notably, the observed changes in pHi in the case of the 2D culture correlated well with cell viability, assessed from the live/dead cell staining (Pearson correlation, *r* = 0.833). In the 3D model, the correlation was moderate, with *r* = 0.622 (Appendix A).

Therefore, the presence of collagen delayed the early decrease in pHi and resulted in a less-pronounced increase in pHi upon prolonged incubation with the drugs of the FOLFOX regimen compared with the cells without collagen.

### 2.4. Metabolic Activity of Cells after Treatment with FOLFOX Regimen

The FLIM of the metabolic cofactor NAD(P)H in untreated HCT116 cells revealed the typical values of the fluorescence decay parameters—the lifetime of the free form (τ_1_) was ~400 ps, the lifetime of the protein-bound form (τ_2_) was ~2600 ps, and the mean τ_m_ was ~794 ps. Thus, the presence of the BC-Ga-Ir probe in the cells did not distort the fluorescence of NAD(P)H.

It was noticed that, upon long-term (>48 h) cultivation in collagen, the HCT116 cells showed a tendency toward a decrease in the NAD(P)H τ_mean_ and an increase in the a_1_/a_2_ ratio, which could be associated with either enhanced glycolysis or reduced mitochondrial respiration (Figure 4). The latter is more likely, because cells growing in collagen have lower proliferation and metabolic rates [38].

FOLFOX treatment caused a gradual increase in the mean fluorescence lifetime of the NAD(P)H (τ_mean_) in both models, without and with collagen, which is an indicator of a shift towards oxidative metabolism. The maximum changes in the cytoplasm were observed after 48 h of incubation with the drugs; however, in the case of the cells cultured without collagen, the statistical differences from the control (852 ps vs. 794 ps, *p* < 0.05) were detected earlier, at 24 h (Figure 4). At the same time, because of the shift to lower lifetimes in the control collagen-embedded cells, the metabolic differences between the treated and untreated populations were more pronounced in this model. The detailed description of how the comparison of the metabolic shift was performed is presented in the Appendix A.

Since the FLIM and PLIM images were obtained from the same fields of view, the correlation between the metabolic state (NAD(P)H τ_mean_) and pH (BC-Ga-Ir τ_m_) was assessed for the individual cells. The Pearson correlation coefficient (r) was 0.356 and 0.199 in the untreated control and FOLFOX-treated cells growing without collagen, correspondingly, which indicated weak associations between these variables (Appendix A). At the same time, there was a good correlation between the effects of the FOLFOX treatment on cellular metabolism and cell viability (Pearson correlation, *r* = 0.773 for 2D and 0.697 for 3D models, Appendix A).

## 3. Discussion

Metabolic and pHi alterations accompany many biological processes and therapeutic responses of cancer cells. In this study, a combined approach that included two-photon excited FLIM and PLIM was applied for the dynamic assessment of metabolic changes and pHi fluctuations in cultured cancer cells upon treatment with the chemotherapeutic regimen FOLFOX. For the first time, to our knowledge, the effects of collagen on the cytotoxicity, metabolism and pHi were revealed for this treatment regimen.

To date, many phosphorescent probes have been developed for various PLIM applications. Since phosphorescence can be effectively quenched by molecular oxygen due to the triplet–triplet energy transfer between the oxygen molecule in the ground state and the triplet-excited state of the probe, most of the phosphorescent probes are oxygen-sensitive and suitable for oxygen assessment in cells and tissues using PLIM [31,33,34]. Several recent works show the possibility of the combined use of PLIM for oxygen measurements and FLIM for metabolic assays based on the autofluorescence of NAD(P)H. For example, in the studies by Kalinina et al. and Parshina et al., using FLIM/PLIM, a correlation was demonstrated between the level of oxygenation and the metabolic status of tumor cells when modeling hypoxia in vitro. In both studies, an increase in the phosphorescence lifetime of oxygen sensors (Ru(BPY)3 or PIr3) and, at the same time, a shift in the metabolic status towards glycolysis was observed under the conditions of a low oxygen content [34,39]. Parshina et al. described the use of the PIr3 oxygen probe to study tumor oxygenation via PLIM in a combination with NAD(P)H FLIM in vivo in mice [34].

In the present work, we performed, for the first time, a simultaneous assessment of pHi using a new phosphorescent probe, BC-GA-Ir, and of the metabolic status using NAD(P)H fluorescence lifetime imaging. Previously, BC-GA-Ir already demonstrated its usefulness as a probe suitable for pHi determination in CHO cells in vitro [27]. Here, we extended the application of BC-GA-Ir to follow intracellular pH in cultured cancer cells upon chemotherapy. While most pH-sensitive dyes are based on fluorescence, the BC-GA-Ir probe is phosphorescent, which is beneficial for imaging in terms of the minimization of background autofluorescence and the possibility of using fluorescence channels for other fluorophores/probes. To the best of our knowledge, there is only one phosphorescent pH probe reported previously. Ma et al. presented a phosphorescent soft salt for the ratiometric and lifetime imaging of intracellular pH variations and demonstrated its performance on the HepG-2 cell line [40].

The decrease in pHi upon FOLFOX treatment is in agreement with previous data, obtained by our and other groups. For example, we demonstrated, both in vitro and in vivo in HeLa cells, expressing a genetically encoded pH sensor, the early acidification of the cytoplasm after treatment with cisplatin [21]. Rebillard et al. reported on intracellular acidification in HT29 cells associated with inhibition of Na^+^/H^+^ membrane exchanger-1 early after cisplatin treatment [41]. The anti-microtubule agent paclitaxel also caused an early, short-time shift to acidic pH in the cytosol [22]. However, the dynamics of the pH changes were different. Here, we observed a prolonged acidification upon treatment, especially in the presence of collagen (up to 24 h). Most likely, this is due to the different intracellular localization of the pH probes in the previous and the present studies. Taking into account the localization of the BC-GA-Ir probe in the Golgi apparatus, we can assume the activation of drug efflux from the cells following the addition of the drugs. It is known that there is an acidic environment in the secretory pathway with the growing acidity from the cis-Golgi to the secretory granules [37]. This assumption is supported by data on the activation of drug efflux through the ABC transporters over the course of chemotherapy [42,43]. The prolonged acidification in cells embedded in collagen may indicate a longer activation of the secretory pathways. Along with the better survivability of cells, it highlights the importance of Golgi apparatus in responsiveness to treatment. However, despite an established activation of ABC transporters and the role of Golgi apparatus in the secretory pathway, data on the direct participation of Golgi apparatus in the efflux of drugs are lacking. So, further investigations of the function of Golgi apparatus in the response to anticancer treatment seem to be promising.

The marked alkalization of the pHi after 48 h was observed only in cancer cells without collagen. Increased pHis were found in both live and dead cells. In the live cells, it can be associated, for example, with apoptosis, which is induced by the treatment [44]. Although apoptosis is often associated with acidic cellular environments, intracellular alkalinization was also demonstrated to be a starting signal for apoptosis, especially in the case of cytotoxic agents [22,45,46,47,48]. In the case of dead cells, the disruption of membrane integrity and the lack of formation of acid metabolites are the most probable reasons for the pHi increase. Previously, we have observed alkaline pHis in the necrotic areas within tumor tissue [49].

In addition to pHi, changes in cellular metabolism upon FOLFOX drug exposure were monitored via the FLIM of NAD(P)H. A shift towards more oxidative metabolism was observed upon treatment in both in vitro models, with and without collagen. In the absence of collagen, similar changes in pHi were observed earlier, at 24 h, compared to 48 h in the collagen-containing samples. Previously, the oxidative shift in chemotherapy-treated cancer cells has been documented in several works, including ours, and it seems to be a nonspecific response to different cytotoxic agents. For example, we have found that chemotherapy with cisplatin causes a decrease in the free NAD(P)H fraction, both in monolayer HeLa cells and tumor xenografts, and this observation correlated with inhibited cancer cell growth. A parallel monitoring of cytosolic pH using the fluorescent protein sensor SypHer2 revealed an early acidification of the cytoplasm in all treated cells [21]. In the paper by Lukina et al., the decrease in the NAD(P)H free/bound (a_1_/a_2_) ratio was observed in colorectal mouse tumors in vivo after treatment with cisplatin, paclitaxel or irinotecan [50]. Our recent study on the HCT116 cell line and patient-derived colorectal cancer cells showed an elongation of the NAD(P)H mean fluorescence lifetime in response to 5-fluorouracil [51]. Therefore, it is not surprising that the FOLFOX combination of drugs produced the same effect. There are several other works that demonstrate an induction of mitochondrial OXPHOS activity in response to chemotherapy by using the FLIM of NAD(P)H [24,38].

Although the combination of PLIM and FLIM provides an opportunity for multiparametric cellular-level imaging, time-resolved data acquisition from live cells imposes certain restrictions on the investigation of dynamic processes. Both the fluorescence emission from NAD(P)H and the phosphorescence emission from the BC-GA-Ir probe are rather weak, so long photon collection times are required to collect a sufficient number of photons for constructing the decay curve, at least 60 s for NAD(P)H and 240 s for BC-GA-Ir. An increase in laser power for more efficient excitation may result in photobleaching and is unsafe for cells. Increases in the concentration of BC-GA-Ir in the case of PLIM can be cytotoxic. Therefore, these parameters—data acquisition time, laser power and concentration of the probe—should be compromised. With the BC-GA-Ir probe, we were able to trace the pHi in the time period from 1 h to 72 h in the same culture dishes as the probe-preserved stable localization within the cells, as confirmed by the microscopic images and unchanged phosphorescent lifetimes in the untreated control. Given the heterogeneity of the intracellular environment and pH in different cell compartments, careful attention should be paid to the temporal and spatial distribution of the chemical probe used in this study.

As for the effects of the tumor extracellular matrix on the efficacy of chemotherapy, the obtained results clearly indicate that the presence of unstructured collagen reduces the therapeutic efficacy of the FOLFOX regimen against colon cancer cells, as assessed from the cell viability, pHi and metabolic assays. The influence of the ECM on tumor responses to chemotherapy is well documented [38,52,53,54,55,56,57,58]. In our recent study on T24 (human urinary bladder carcinoma) cells growing in 3D collagen-based models, we showed that collagen protects cancer cells from the action of doxorubicin, improves viability and diminishes cellular metabolic changes upon treatment [38]. In the paper by Chen et al., a low TACS (Tumor-Associated Collagen Signature, the index that positively correlates with collagen area, collagen straightness and collagen cross-link density) level in gastric tumors was associated with improved outcomes in patients who received 5-fluorouracil-based adjuvant chemotherapy [55]. Similar results were obtained for patients with gastric cancer by Yang et al. [56]. At the same time, cancer cells in unorganized collagen were more sensitive to a 5-fluorouracil and FOLFOX combination than the cells within an organized stroma in co-culture with hepatic stellate cells, as was shown on the model of tumor spheroids in Ref. [57]. However, in this work, the control without the extracellular matrix was not presented.

The retrospective analysis of clinical data from metastatic colorectal cancer patients who received renin–angiotensin system inhibitors (RASIs) during combined anti-angiogenic and chemotherapeutic treatment showed significant survival benefits of overall survival and progression-free survival over patients who did not receive RASIs. Subsequent in vivo studies have demonstrated that RASIs inhibit collagen and hyaluronic acid deposition [58]. The increased content of fibronectin, one of the major extracellular components, was shown to be associated with the chemoresistance of colorectal cancer [59]. In the study by D’Angelo et al., cancer cells grown in a scaffold from colorectal cancer liver metastasis showed increased resistance to chemotherapy [60].

Therefore, most of the research demonstrates that either separate components of the extracellular matrix or their combination have a protective effect on cancer cells under chemotherapy, which is in good agreement with our results.

A limitation of our study is that only unstructured collagen was used in a 3D model. Although some cancer cell lines are able to remodel collagen, in the case of HCT116 cells, fibrillar collagen was present in a low amount [61]. One of the ways to structure collagen is to add fibroblasts in the collagen hydrogel, but this can impact the biological behavior of cancer cells. An investigation of the effects of collagen structure on tumor cells’ metabolism and sensitivity to chemotherapy will be a subject of our further research.

## 4. Materials and Methods

### 4.1. Cell Cultures and Models

Human colorectal cancer cell line HCT116 was obtained from the Cell Culture Collection of the Ivanovsky Institute of Virology, Gamaleya National Research Center of Epidemiology and Microbiology (Moscow, Russia). Cells were cultivated in Dulbecco’s modified Eagle’s medium (DMEM; PanEco, Moscow, Russia) supplemented with 10% fetal bovine serum (FBS) (PanEco, Moscow, Russia), 2 mM glutamine (PanEco, Moscow, Russia), 10 mg/mL of penicillin and 10 mg/mL of streptomycin at 37 °C, 5% CO_2_ and 80% relative humidity in the CO_2_ incubator SCO2W (Shel Lab, Cornelius, OR, USA). The cells were routinely passaged twice a week using 0.025% trypsin-EDTA (PanEco, Moscow, Russia).

Studies were carried out on the cells grown either as a monolayer (2D) or as a three-dimensional (3D) model in collagen. A schematic of the cellular models used in this study is presented in Figure 5. For the monolayers, the cells were seeded in 35 mm glass-bottomed FluoroDishes (Ibidi GmbH, Gräfelfing, Germany) at an amount of 3 × 10^5^ cells in 2 mL of DMEM, and incubated for 24 h (37 °C, 5% CO_2_). Then, the cells were washed with phosphate-buffered saline (PBS) and placed in a FluoroBright DMEM (Thermo Fisher Scientific, Waltham, MA, USA) containing 10% FBS. The 3D model was obtained using a previously developed protocol [61]. Briefly, a solution of type I rat tail collagen (1.5 mg/mL) was mixed with a reagent mixture in the volume ratio of 3.5:1. The reagent mixture for gel neutralization was 10× Medium 199 (Gibco, Life Technologies, Carlsbad, CA, USA), NaOH, Na_2_CO_3_, glutamine and 1× HEPES (volume ratio of reagents: 16:8:25:1:5). Then, a suspension of HCT116 cells was added to the collagen gel in the ratio of 1:10. The total cell concentration was 3 × 10^5^ cells/mL, with a final concentration of collagen gel of 1.0 mg/mL.

### 4.2. MTT Assay and In Vitro Chemotherapy

To select drug concentrations for the FOLFOX regimen, seven combinations of the drugs were tested using an MTT assay (Table 1). The cells were seeded in 96-well plates (5 × 10^3^ cells per well) and incubated for 24 h. Then, combinations of the drugs oxaliplatin (Teva, Haarlem, The Netherlands), 5-fluorouracil (VeroPharm, Volginsky, Russia) and leucovorin (Teva, Haarlem, The Netherlands) in the ratio of 1:2:4, corresponding to the FOLFOX regimen, were added. After 72 h of incubation, the cells were treated with the MTT reagent 3(4,5-dimethyl-2-thiazolyl)-2,5-diphenyl-2H-tetrasolebromide (PanEco, Moscow, Russia) according to the manufacturer’s protocol, and the colorimetric analysis was performed at a wavelength of 570 nm using a multimode microplate reader (Synergy Mx; BioTek Instruments, Winooski, VT, USA). The cell viability was calculated as the percentage of untreated control cells. The experiment was performed three times with 8–10 internal replicates for each dose. The MTT curve is presented in Figure 6.

Further treatments of the cells were performed using concentrations of oxaliplatin (14 μM), 5-fluorouracil (56 μM) and leucovorin (28 μM), which provided ~50% cell viability. 

### 4.3. pH Probe and Staining Protocol

The pH-sensitive BC-Ga-Ir probe was prepared according to [27]. The probe was a cross-linked block-copolymer of vinylppirolidone-vinylamine conjugated with an iridium complex. The pH-sensitive probe consisted of a luminescent bis-cyclometalated iridium(III) complex containing a pH-sensitive carboxylic group that gave emission parameter responses to variations in media acidity. The complex was embedded into the block-copolymer nanospecies based on poly(N-vinyl pirrolydone-block-N-vinyl amine), which was additionally cross-linked with glutaraldehyde, to eliminate the quenching effect of oxygen and the influence of the components of biological systems on the sensor emission characteristics. The abbreviation “BC-Ga-Ir” comes from the description of the sensor components: BC means “block-copolymer”, GA is glutaraldehyde used as a cross-linking agent and Ir is the iridium complex. The hydrodynamic diameter of the probe was 31 nm. The absorption maxima of the probe were at 261, 338 and 450 nm; the emission band was centered at 640 nm. BC-Ga-Ir had a pKa ≈ 6.5, with sensitivity to pH in the range from 4 to 9 pH, which fully covers physiological pH values. The phosphorescence lifetime of the probe shows a linear dependence on pH. In a model solution of DMEM with FBS (99:1 vol.%, respectively), the lifetime increased from 0.6 to 1.4 μs, with an increase in pH from 5.1 to 8.2 units.

The working solution (0.5 mg/mL) was prepared in FluoroBrite DMEM immediately before the experiment; then, it was mixed with the FOLFOX combination of drugs and added (2 mL per culture dish) to the cells 24 h after cell seeding. The working solution without drugs was used for the control measurements.

To determine the subcellular localization of the probe, colocalization experiments were performed using organelle-specific dyes: LysoTracker™Yellow HCK-123 (Invitrogen, Waltham, MA, USA) for the identification of lysosomes and ER-Tracker™ Green (BODIPY™ FL Glibenclamide) (Invitrogen, USA) for the identification of the endoplasmic reticulum (ER) and Golgi apparatus (GA). The cells in the amount of 10 × 10^3^ were seeded in a 96-well plate for confocal microscopy (Corning, Glendale, AZ, USA). Trackers were used according to the manufacturer’s protocol. The LysoTracker was dissolved in FluoroBrite DMEM (final concentration: 50 nM); the ER-Tracker™ was dissolved in Hank’s solution (final concentration: 1 μM). The cells were incubated with the BC-Ga-Ir probe for 24 h, and then the trackers were added. The cells were incubated with the trackers for 1 h. Using the microscope LSM 880 (Carl Zeiss, Jena, Germany), a simultaneous registration of the signals from the BC-Ga-Ir probe and the tracker was performed. The following settings were used: excitation at 405 nm and registration of the signal in the 550–650 nm range for the BC-Ga-Ir probe; excitation at 488 and registration in the 500–550 nm range for the ER-Tracker™; excitation at 488 and registration in the 550–650 nm range for the LysoTracker™. Separate wells were used for each tracker.

### 4.4. Phosphorescence Lifetime Imaging Microscopy (PLIM)

The experiments were performed on a laser scanning microscope, the LSM 880 (Carl Zeiss, Jena, Germany), with a PLIM option: a time-correlated single photon counting card (SPC-150), a hybrid detector (HPM-100–40), a digital Delay Generator card (DDG-210) (Becker & Hickl GmbH, Berlin, Germany) and INDIMO PLIM (Carl Zeiss, Jena, Germany). The phosphorescence of the BC-Ga-Ir probe was excited in the two-photon mode with a femtosecond Ti:Sa laser, the Mai Tai HP (Spectra-Physics, Milpitas, CA, USA, 80 MHz, 140 fs), at a wavelength of 810 nm and registered in the range of 495–690 nm. The phosphorescence signal was collected for 240 s. The number of photons per pixel was 600–1000 photons. During the experiment, the cells were in an XL multi S Dark LS incubator (PeCon GmbH, Donau, Germany) at 37 °C and 5% CO_2_. The experiments were carried out in a dark room with the detectors isolated from ambient light in order to increase the accuracy of measuring the phosphorescence lifetime and reduce the noise contribution.

The PLIM images were processed using the SPCImage 8.5 software (Becker & Hickl, Berlin, Germany). For fitting the obtained decay curves, the number of photons was increased to 5000 using the binning option. An algorithm of the incomplete decay curve was used. The phosphorescence decay curves were fitted by a single-exponential model; only the images with an appropriate χ^2^ value (from 0.8 to 1.2) were processed. The phosphorescence lifetimes were calculated in the individual cells by selecting the whole cell as region of interest. Calculations were performed in 5–7 microscopic fields of view, with a total number of cells of 25 to 50 for each time point.

Calibration of the lifetime vs. pH was performed on cells in vitro using a set of buffer solutions containing 130 mM KGluconate, 2 mM CaCl_2_, 1 mM MgCl_2_, 10 μM nigericin and 30 mM MOPS or TRIS with pH values of 6.0, 7.0, 9.0 [17]. The pH was adjusted with 1 M KOH or 1 M HCl at room temperature (the registration of phosphorescence was also performed at room temperature). The cells were incubated with a buffer for 30 min, and then the phosphorescence was recorded as described above. For each pH value, the phosphorescence signal from 5 fields of view was acquired. The obtained dependence was nearly linear. The phosphorescence lifetime values were converted into pH using the following equation: pH = 4 + (τ_m_ − 1.056)/0.085.

### 4.5. Fluorescence Lifetime Imaging Microscopy (FLIM)

The fluorescence of the metabolic cofactor NAD(P)H was recorded using an LSM 880 laser scanning microscope (Carl Zeiss, Jena, Germany) with a TCSPC-based FLIM module (Becker & Hickl GmbH, Berlin, Germany). The fluorescence of NAD(P)H was excited at a wavelength of 750 nm and detected in the range of 450–490 nm. The laser power on the sample was 6 mW. The photon collection time was 60 s, which allowed us to collect >3000 photons per decay curve at a binning of 1. The FLIM images were obtained sequentially from the same fields of view as the PLIM images.

In the 3D model, the FLIM and PLIM images were captured from the depth of ~10 µm. In a preliminary study, the Z-stack of FLIM images of NAD(P)H was obtained up to the depth of 123 μm (Figure 5C). However, it was noticed that at a depth of >25 µm, the fluorescence intensity of NAD(P)H decreased, allowing only 1000 photons or less to be collected per decay curve, which is insufficient for the bi-exponential fitting. Therefore, all further measurements were performed from the cell layers closest to the surface of the glass bottom, where the number of photons was optimal for the FLIM data processing.

To obtain statistics on the FLIM parameters for individual cells, we performed an automatic analysis of the images based on the corresponding regions of interest. To analyze individual cells, we used a segmentation algorithm previously described in [51]. Briefly, the segmentation procedure involved delineating cell boundaries and cell nuclei using the U-net neural network, followed by segmentation based on a watershed algorithm. The U-net model was trained on images of human colorectal adenocarcinoma cells [51] using the Dice loss function. The U-net model implementation was carried out in the Keras library in Python 3.1, and the image processing libraries skimage and scipy were used.

Next, the fluorescence decay curves of NAD(P)H for individual cells were approximated using a standard biexponential decay model [26] using scripts in Python 3.1 with the IMFIT library. The characteristic decay times (τ_1_ and τ_2_, respectively) and their relative amplitudes (a_1_ and a_2_, where a_1_ + a_2_ = 100%) were obtained, as well as the mean fluorescence lifetime (τ_mean_ = (a_1_ · τ_1_ + a_2_ · τ_2_)/(a_1_ + a_2_)). The fitting quality, represented by the χ^2^ value, ranged from 0.7 to 1.3. In a first approximation, the first (short, τ_1_) component of NAD(P)H fluorescence decay is attributed to its free state associated with glycolysis, while the second (long, τ_2_) component is associated with the protein-bound state related to the mitochondrial respiratory chain. Thus, the statistics of the FLIM parameters of the cytoplasm of each cell was obtained, about 500 cells for each time point.

### 4.6. Second Harmonic Generation (SHG) Microscopy

To visualize fibrillar collagen in 3D collagen hydrogel, the SHG imaging option on the LSM880 microscope was used. The following parameters were used: excitation at a wavelength of 800 nm and detection in the range of 371–421 nm in the backward direction. The average laser power was ~12 mW.

### 4.7. Colony-Forming Assay

The HCT116 cells were seeded on 6-well plates in a concentration of 3 × 10^5^ cells per well in a complete growth medium or in collagen hydrogel. After 24 h, the drugs of the FOLFOX combination were added in the IC50 concentration. In 24 h of incubation with the drugs, the number of live HCT116 cells were counted with trypan blue exclusion using an automated cell counter T20 (Bio-Rad, Hercules, CA, USA). Then, live cells in the amount of 4000 cells per well were plated in triplicate into six-well plates in DMEM supplemented with FBS. Colonies formed after 10 days were fixed with ice-cold alcohol (95.0%), stained with crystal violet (0.5%, 500 mL per well) and observed using a Leica DMIL microscope (Leica, Wetzlar, Germany). For quantitative analysis, the number of colonies in each well was counted; then, the crystal violet was eluted with alcohol (1 mL per well) and the optical density was measured at 570 nm using a multimode microplate reader (Synergy Mx; Bio-Tek Instruments, Winooski, VT, USA).

### 4.8. Live/Dead Cell Assay

The cell viability was assessed with calcein and propidium iodide (PI) using a live/dead cell double staining kit (Sigma, St. Louis, MO, USA) according to the manufacturer’s protocol. The HCT116 cells were seeded in a 6-well plate with or without collagen and treated with the FOLFOX regimen. The cells were stained for 1, 5, 24, 48 and 72 h; then, the PI-stained (dead cells) and calcein-stained (live cells) areas in each field of view were calculated for the control and treated cells. Fluorescence images were obtained using a wide-field Leica DMIL fluorescence microscope with a YFP ET filter (Ex: BP 500/20, Em: BP 535/30) for calcein and a TX2 green filter (Ex: BP 560/40, Em: BP 645/75) for PI.

### 4.9. Statistical Analysis

The PLIM data were tested for normal distribution using the Kolmogorov–Smirnov test. Non-normally distributed data are presented as the median and Q1 and Q3. The Wilcoxon *t*-test was used to compare data, with *p* < 0.05 considered statistically significant. Statistical data processing was carried out using the RStudio 9.3.191259 software (Boston, MA, USA).

For the colony-forming assay and live/dead cell assay, the mean values (*n* = 5) and standard deviations (SD) were calculated.

## 5. Conclusions

The ECM’s distribution and composition within tumors is highly heterogeneous. Therefore, the effects of the ECM on the biological behavior of cancer cells and their responsiveness to therapies can vary. Our study shows that the presence of unstructured collagen in the microenvironment of cancer cells results in a lower efficacy of the chemotherapeutic FOLFOX regimen, and the changes in cellular metabolism and pHi that accompany cellular-level therapeutic responses are less pronounced. Moreover, collagen drives the metabolic rearrangements towards glycolysis, which is beneficial for cancer cell survival. A multiparameter, quantitative, non-invasive analysis of cellular processes became possible, owing to the combined use of time-resolved microscopy techniques—PLIM for pHi assessment and FLIM for NAD(P)H-based metabolic imaging. Taking into account a variety of stroma architectures within tumors, further studies are needed to uncover the associations of therapeutic effects with the ECM structure.

## Figures and Tables

**Figure 1 ijms-25-00049-f001:**
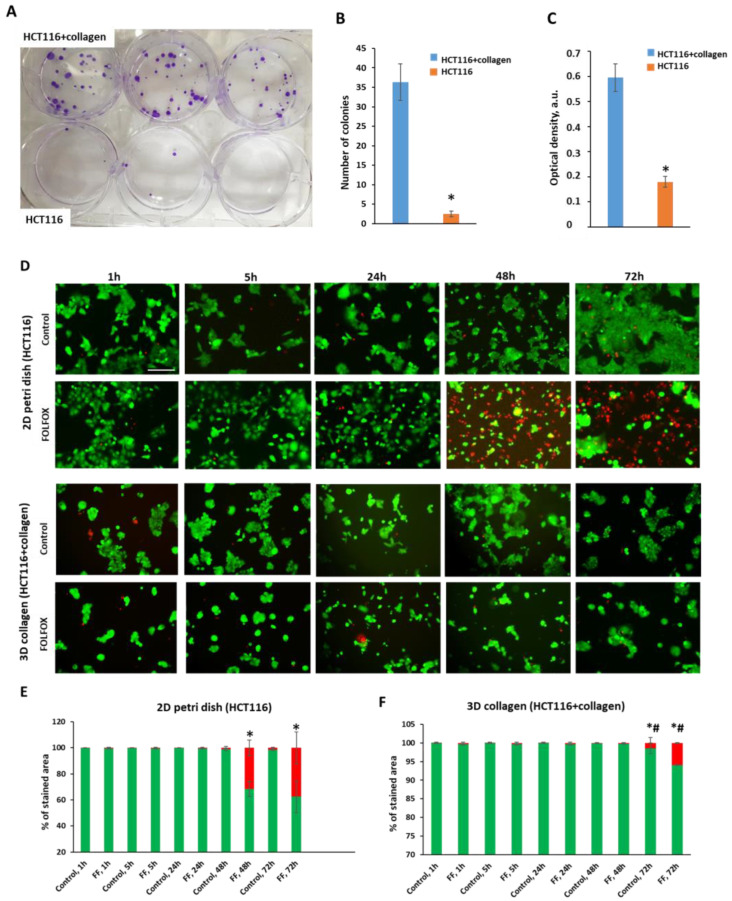
Effects of collagen on cytotoxicity of FOLFOX regimen in HCT116 cancer cells. (**A**) Colony-forming assay after treatment in the presence and absence of collagen. Representative photographs of the culture dishes. (**B**) The number of colonies formed by HCT116 cells after FOLFOX treatment. (**C**) Optical density of the eluted crystal violet dye. Mean ± SD. *, *p* = 0.000001 HCT116 vs. HCT116 + collagen. (**D**) Live/dead cell assay after 24 h and 48 h exposures to FOLFOX in the absence (**upper**) and presence of collagen (**lower**). Fluorescence microscopy images of the stained cells: green—live cells (calcein), red—dead cells (propidium iodide). The scale bar is 200 µm for all images. (**E**) Quantification of the areas stained with propidium iodide (red columns) and calcein (green columns) in a monolayer of HCT116 cells. (**F**) Quantification of the areas stained with propidium iodide (red columns) and calcein (green columns) in HCT116 cell culture in collagen. Mean ± SD. *, *p* = 0.000001 HCT116 vs. HCT116, control 1 h, #, *p* = 0.000001 HCT116 vs. HCT116 + collagen at the same timepoint. “Control” are the cells without treatment, “FF” are the cells treated with FOLFOX.

**Figure 2 ijms-25-00049-f002:**
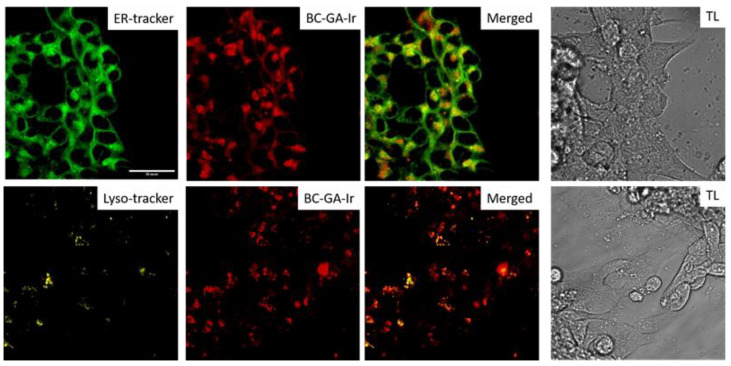
Colocalization analysis of endoplasmic reticulum/Golgi apparatus, lysosomes and BS-Ga-Ir probe. For ER/AG: ex. 488 nm, reg. 500–550 nm; for lysotracker: ex. 488 nm, reg. 550–650 nm; for BC-GA-Ir: ex. 405 nm, reg. 550–650 nm. One-hour incubation with tracers. Scale bar is 30 μm.

**Figure 3 ijms-25-00049-f003:**
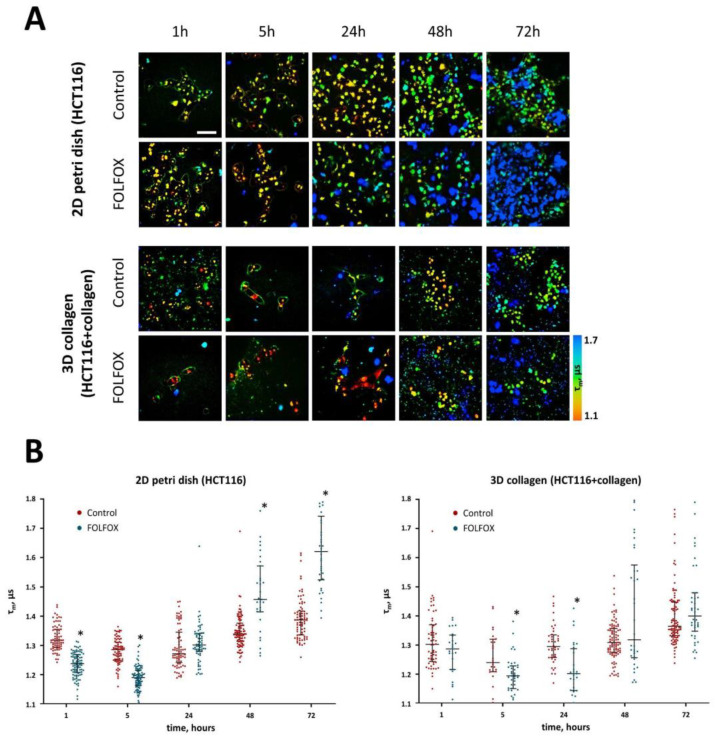
Changes in intracellular pH induced by FOLFOX regimen in HCT116 cancer cells cultured in the absence and in the presence of collagen. (**A**) PLIM images of cells stained with the BC-Ga-Ir probe over the course of incubation with the drugs. Controls are the cells without treatment. Scale bar = 50 μm. (**B**) Phosphorescence lifetimes of BC-GA-Ir in control and treated cancer cells. The median and the quartiles Q1 and Q3 are shown. Dots are the measurements in the individual cells. *n* = 25–50 cells. * *p* ≤ 0.05 vs. controls.

**Figure 4 ijms-25-00049-f004:**
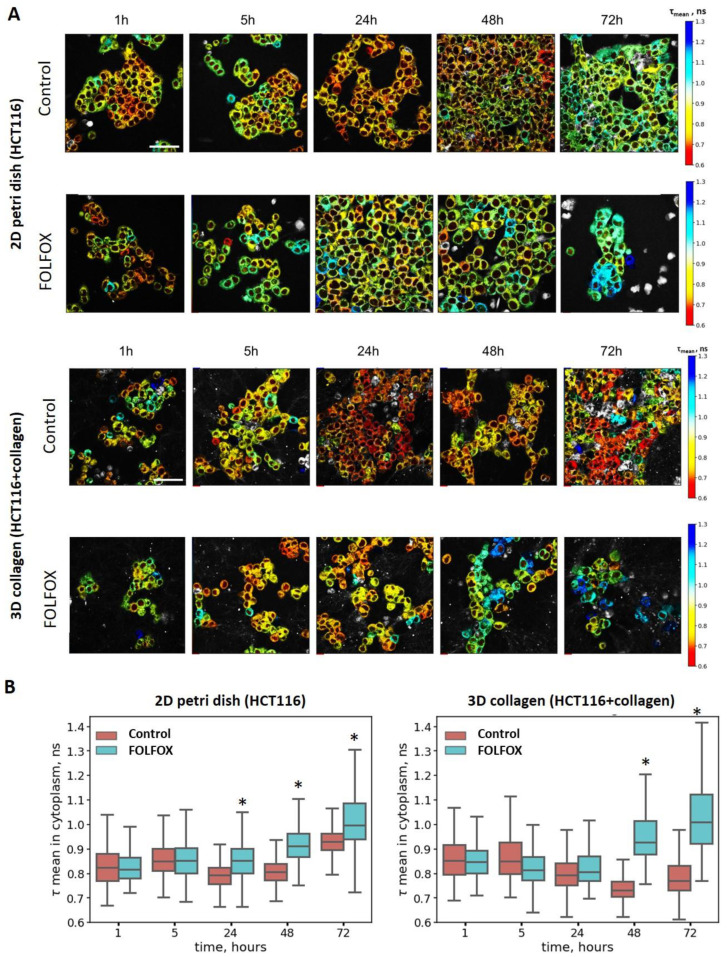
Metabolic changes induced by FOLFOX regimen in HCT116 cancer cells cultured in the absence and in the presence of collagen. (**A**) NAD(P)H FLIM images over the course of incubation with the drugs. Controls are the cells without treatment. Scale bar = 50 μm. (**B**) Analysis of the mean fluorescence lifetime of NAD(P)H in the cytoplasm of control and treated cells. Box shows the median and the quartiles Q1 and Q3; whiskers are minimum and maximum. *n* = 500 cells. * *p* ≤ 0.05 vs. controls.

**Figure 5 ijms-25-00049-f005:**
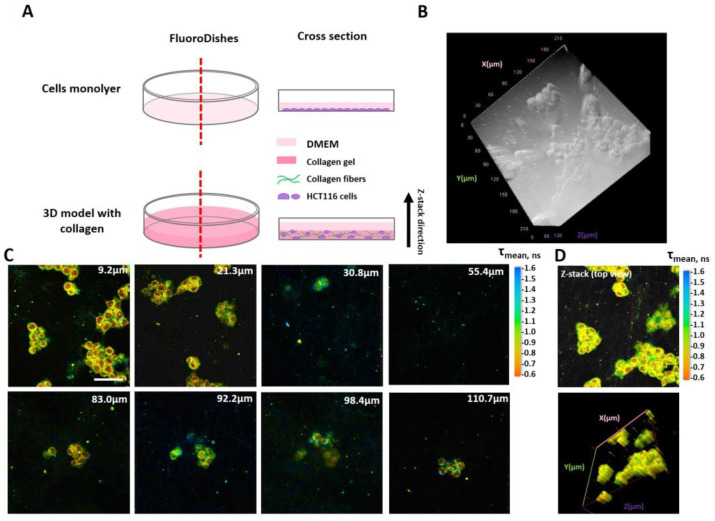
Cellular models used in this study: 2D monolayer and 3D collagen hydrogel. (**A**) The schematic of the models. (**B**) Three-dimensional transmitted light image of cancer cells in collagen hydrogel. (**C**) NAD(P)H FLIM images obtained from the indicated depth in a 3D model (excitation: 750 nm, detection: 450–490 nm). The mean fluorescence lifetime (τ_mean_) is color-coded. Scale bar = 50 μm for all images. (**D**) Confocal Z-stack of NAD(P)H FLIM images combined from all focal planes. The 3D image was acquired at 24 h after cell seeding In collagen gel. The microscope’s magnification is 40×.

**Figure 6 ijms-25-00049-f006:**
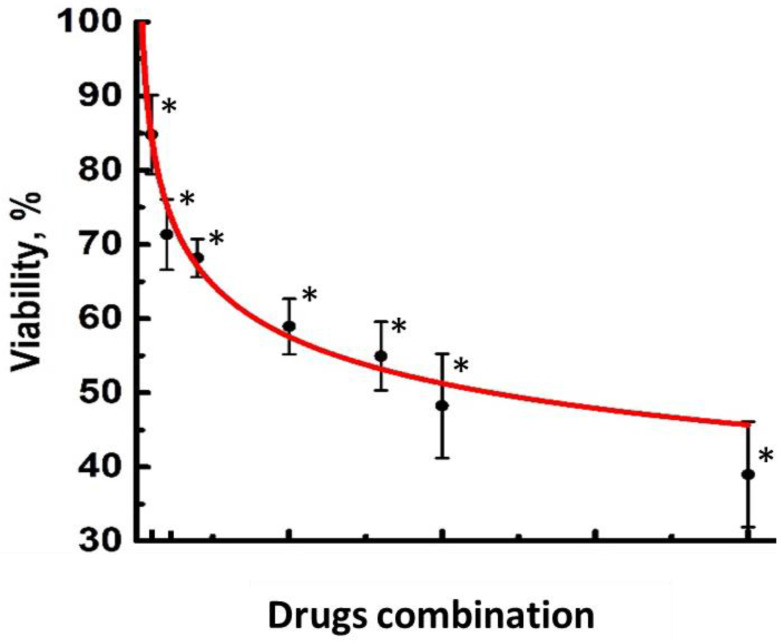
MTT assay for HCT116 cell viability in the presence of FOLFOX combinations of the drugs. Mean ± SD, *n* = 8–10. * *p* = 10^−6^ vs. control.

**Table 1 ijms-25-00049-t001:** Drug combinations tested on HCT116 cell line.

Combination	1	2	3	4	5	6	7
Oxaliplatin, μM	0.5	1	2	5	8	10	20
Leucovorin, μM	1	2	4	8	16	20	40
5-Fluorouracil, μM	2	4	7	14	25	35	70

## Data Availability

The data presented in this study are available on request from the corresponding author.

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
