# Peer review of "Monitoring the Intracellular pH and Metabolic State of Cancer Cells in Response to Chemotherapy Using a Combination of Phosphorescence Lifetime Imaging Microscopy and Fluorescence Lifetime Imaging Microscopy"

_ijms, 2023, doi:10.3390/ijms25010049_

Round 1
Reviewer 1 Report
Comments and Suggestions for Authors
Tunik et al. reported the combined techniques of PLIM and FLIM for simultaneous monitoring of pHi and metabolic state (NAD(P)H), respectively, and the association between pHi and this metabolism under chemotherapy that employed FOLFOX. The author use 3D collagen based hydrogel as the ECM culture model. Their results indicated that chemotherapy caused early temporal decrease of pHi and then increase of pHi, along with the shifting of glycolysis into oxidative respiration. These results are highly interesting and insightful to understand the biochemical changes inside the cells when treated by chemotherapy. The findings provide novel insights into molecular targets for cancer cells. The following comments can be addressed before publication:
1. It is known that ECM stiffnesses can also influence the intracellular signalings (e.g., cell stemness and malignancy) of cancer cells through biomechanical forces apart from biochemical cues. Relevant literature can be cited and discuss in the introduction part (e.g., Sci. Adv., 2023, 9 (27), eadg9593; Cancer Res. 2011, 71(15):5075-80)
2. What does the colors represent in Figure 1C? Seems to be unclear and what is the time point of the culture? The purpose of figure 1B is also unclear and the 3D effect is not obvious. The scale of the color bar is also unclear.
3. The naming for each group in Figure 2 seems inappropriate. (1) HCT116 and (2) HCT116+collagen. E.g., (1) 2D petri dish and (2) 3D collagen are better naming to avoid confusion. What do control and FF represent in 1B? Figure 2 legend “p=0.000 HCT116 vs HCT116+collagen” why p = 0? The green and red colors in 1B should be calcein and PI, right? They need labels.
4. What does colocalization of BC-GA-Ir and ER/Golgi imply?
5. “In the treated groups, the number of cells with long lifetime was notably higher, and at long incubation times they presented the majority in the cell population.”
6. I think the author can try to present data of the subpopulation with various pHi, such as dotted plotting for distribution of individual cells in Figure 4.
7. The naming for the groups in Figure 4-5 is also quite confusing. What is the control for HCT116+collagen? Culture on the surface?
8. Is there any analysis/correlation between the results of PLIM and FLIM?
Comments on the Quality of English Language
Can be further polished
Author Response
- It is known that ECM stiffnesses can also influence the intracellular signalings (e.g., cell stemness and malignancy) of cancer cells through biomechanical forces apart from biochemical cues. Relevant literature can be cited and discuss in the introduction part (e.g., Sci. Adv., 2023, 9 (27), eadg9593; Cancer Res. 2011, 71(15):5075-80)
Thank you for your recommendation, we added this information to the Introduction part.
- What does the colors represent in Figure 1C? Seems to be unclear and what is the time point of the culture? The purpose of figure 1B is also unclear and the 3D effect is not obvious. The scale of the color bar is also unclear.
Thank you for your remarks. The colors in the in Figure 1C represent the values of the mean fluorescence lifetime (τmeam) of NAD(P)H, the detailed description of this parameter can be found in the Material and methods section “Fluorescence lifetime imaging microscopy (FLIM)”. We changed the color bar to make it clear. Images were acquired in 24 hours after cells seeding. The figure 1B was included to show the distribution of cells in collagen hydrogel. As long as collagen has a rather dense structure, a lower amount of cells can be seen in the field of view in comparison with 2-D culture.
- The naming for each group in Figure 2 seems inappropriate. (1) HCT116 and (2) HCT116+collagen. E.g., (1) 2D petri dish and (2) 3D collagen are better naming to avoid confusion. What do control and FF represent in 1B? Figure 2 legend “p=0.000 HCT116 vs HCT116+collagen” why p = 0? The green and red colors in 1B should be calcein and PI, right? They need labels.
We made required corrections in the Figure 2 and the legend.
- What does colocalization of BC-GA-Ir and ER/Golgi imply?
We added an explanation in the text of the article. “Colocalization experiments clearly indicated that BC-GA-Ir probe is localized mainly in Golgi apparatus and endoplasmic reticulum (the Manders overlap coefficient, M1 0.855) and partly in the lysosomes (M1 0.594) (Fig. 3), which means that pH is measured mainly in these compartments.”
- “In the treated groups, the number of cells with long lifetime was notably higher, and at long incubation times they presented the majority in the cell population.”I think the author can try to present data of the subpopulation with various pHi, such as dotted plotting for distribution of individual cells in Figure 4.
We thank the Reviewer for this comment. The diagram was changed and presented as dot-plots.
- The naming for the groups in Figure 4-5 is also quite confusing. What is the control for HCT116+collagen? Culture on the surface?
We thank the Reviewer for this recommendation. We changed naming to make it clear.
- Is there any analysis/correlation between the results of PLIM and FLIM?
We thank the Reviewer for the valuable comment. We have performed correlation analysis for the individual cells. The results are included in the Manuscript.
Figure S5. Analysis of correlation between metabolic state (NAD(P)H τmean) and pH (BС-Ga-Ir τm) at the single-cell level. Scatter plots for untreated control (A) and FOLFOX treated cells (B). Dots are the measurements for individual cancer cells in a 2D model. Pearson’s correlation coefficients r are indicated on each plot. The solid red line represents the regression line.
Reviewer 2 Report
Comments and Suggestions for Authors
Reviewer’s comments on manuscript „Monitoring of intracellular pH and metabolic state of cancer cells in response to chemotherapy using a combination of PLIM and FLIM (#IJMS-2708577)” by I.N. Druzhkova et al.
This is a well written manuscript about the opposing effect of the extracelluler matrix of colorectal carcinoma cells excerted against FOLFOX chemoterapy. Mitigating effects of extracellular millieu, represented here by collagen, on FOLFOX induced changes in intracellular pH and metabolism, as monitored by PLIM utilising the BC-Ga-Ir phosphore as the pH indicator and by FLIM utilising NAD(P)H fluorophore as the metabolism indicator has been investigated. The paper is important, because it might contribute to a better curing of colorectal cancer. Additionally by combining phosphorescence and fluorescence lifetime measurements it also nicely demonstrates an effective and rather global approach of the problem. The experiments are well done with all the necessary controls. I noticed only some typographical errors, which I list below.
Formal errors:
1. In the Legend for Figure 2 the „p=0.000” is misleading, or at least it’s meaning is not clear at the first sight. Appending a verbal explanation is suggested as a completion.
2. In all the figures where it applies, please replace the „FF” with ”FOLFOX”, for the easier understanding.
3. In the designation of the y-axes in Panel B of Figure 4, please write microsec (micros, ms) instead of ms.

Author Response
Formal errors:
- In the Legend for Figure 2 the „p=0.000” is misleading, or at least it’s meaning is not clear at the first sight. Appending a verbal explanation is suggested as a completion.
We thank the Reviewer for this comment. The Figure and legend were changed.
- In all the figures where it applies, please replace the „FF” with ”FOLFOX”, for the easier understanding.
We thank the Reviewer for this comment. The Figure and labels were changed.
- In the designation of the y-axes in Panel B of Figure 4, please write microsec (micros, ms) instead of ms.
Unfortunately, we can not agree with this comment. The unit name “µs” was used as a standard abbreviation for microseconds.
Reviewer 3 Report
Comments and Suggestions for Authors
The article is interesting and the authors in the manuscript have addressed a current research topic concerning the extracellular matrix and the role of collagen, which is one of the factors in the heterogeneous response of cancer cells to chemotherapy and a strong regulator of their metabolic behaviour. In the manuscript, the introduction is appropriately specified and the purpose of the research is clear. The discussion contains a broad discussion of the problem.
However, there are shortcomings affecting the quality of the manuscript. The materials and methods were not written carefully, and the presentation of the research results requires some modification.
General note. The manuscript was not prepared based on IJMS guidelines. You should use the template available on the magazine's website. The numbering of each line is missing, making it difficult to review. No data on corresponding author, chapter numbering, etc.
Below are the comments that need to be improved:
1. Keywords. I think you should consider changing your keywords as the current ones are too general. I propose to replace: cancer cells to HСT116 cell line; chemotherapy to FOLFOX.
2. Materials and methods.
2.1. Cell cultures and models.
Must be completed:
-where did the cell line used come from,
- cell culture incubator model,
-volumes of reagents used to form 3D collagen hydrogel,
In Figure 1C, in addition to the layers made in the Z axis, a image created by collecting the scanned focal layers should be shown.
2.2. In vitro chemotherapy. I propose to transfer the data content from Supplementary materials to the manuscript, especially the table presenting the range of concentrations of the tested drugs (FOLFOX treatment regimen). The table should be prepared using a Microsoft Word template. This part also lacks information about the incubation time of the tested drugs with the cells. From the discussion of the MTT test, it appears that the test was performed after 72 hours. However, other experiments presented in the manuscript were conducted after 24 and 48 hours. This needs to be standardised.
I also propose to include the methodology for the MTT test in the Materials and Methods, and the MTT results in the Results section. Significance was not indicated in the cell viability graph, and the manuscript lacks information about the statistical test used for MTT. I think it will be more readable and will better show the order of experiments performed.
Table S1 shows 7 drug combinations, and Figure S1 shows 20 combinations (inconsistency). The table also lacks the concentration combinations of drugs used in the studies. The Y axis should show % of viability. The control group should be taken as 100%. A detailed methodology for the MTT test should also be provided. Error in the name MTT (should be .....-thiazolyl).
2.3. pH-probe and staining protocol/Colony-Forming Assay. The volume of tracers and reagents used is missing.
2.4. Live/Dead Cell Assay. Information on the number of cells counted is missing.
3.Results.
3.1. Effects of Collagen on Cytotoxicity of FOLFOX.
Figure 2A. The colony-forming assay. According to the assumption of the test, the colonies of surviving cells should be counted and a cell survival curve should be plotted. Only the optical density of crystal violet is given in the figure.
Figure 2B. In the discussion of the results, the authors write about the change in the number of live and dead cells. However, no numerical or percentage values were given.
Figure 3. The figure caption lacks information about the cell incubation time during which dyes colocalization was observed.
Figures 4 and 5. The figures show additional incubation times of 1 h and 5 h in addition to 24, 24 and 72 h. What were these times used for? The background in the presented photographs should also be reduced (cut off) (Figure 4A).
In summary, the conducted research is interesting, but the manuscript currently requires modifications, especially in terms of graphical presentation of the results.
Author Response
- I think you should consider changing your keywords as the current ones are too general. I propose to replace: cancer cells to HСT116 cell line; chemotherapy to FOLFOX.
Thank you for your recommendation. We changed the keywords
- Materials and methods.
2.1. Cell cultures and models.
Must be completed:
-where did the cell line used come from,
- cell culture incubator model,
-volumes of reagents used to form 3D collagen hydrogel,
Thank you for your recommendations. We added necessary information to the text
In Figure 1C, in addition to the layers made in the Z axis, a image created by collecting the scanned focal layers should be shown.
Thank you for your recommendation. We added FLIM-Z-stack image in Figure 1.
2.2. In vitro chemotherapy. I propose to transfer the data content from Supplementary materials to the manuscript, especially the table presenting the range of concentrations of the tested drugs (FOLFOX treatment regimen). The table should be prepared using a Microsoft Word template. This part also lacks information about the incubation time of the tested drugs with the cells. From the discussion of the MTT test, it appears that the test was performed after 72 hours. However, other experiments presented in the manuscript were conducted after 24 and 48 hours. This needs to be standardised.
The part of the text related to the MTT assay was modified. For better understanding we included the MTT assay for oxaliplatin, the main drug in the therapeutic FOLFOX regimen. The concentrations of two other drugs were calculated according to the ratio of 4:1:2 (5-fluorouracil : oxaliplatin : leucovorin). We performed MTT-assay in 72 h of incubation with the drug as it is a “gold standard” for determination of the IC50 value. The experiments on pH and metabolism monitoring were performed at 1h, 5h, 24h, 48h, and 72h to investigate both the early and late effects of the treatment. For verification of therapeutic effects of FOLFOX, additional methods were used at 24 h and 48 h (colony-forming assay, live-dead cell assay), which are applicable for both models, 2D cell monolayer and 3D collagen hydrogel, unlike the MTT assay.
I also propose to include the methodology for the MTT test in the Materials and Methods, and the MTT results in the Results section. Significance was not indicated in the cell viability graph, and the manuscript lacks information about the statistical test used for MTT. I think it will be more readable and will better show the order of experiments performed.
We agree with the Reviewer. We included the MTT-assay in the Materials and Methods and Results. Statistical analysis for the MTT assay was added in the Supplementary materials.
Table S1 shows 7 drug combinations, and Figure S1 shows 20 combinations (inconsistency). The table also lacks the concentration combinations of drugs used in the studies. The Y axis should show % of viability. The control group should be taken as 100%. A detailed methodology for the MTT test should also be provided. Error in the name MTT (should be .....-thiazolyl).
We agree with the Reviewer and thank the Reviewer for the valuable comments. We have made the corresponding changes in the text and Figure.
2.3. pH-probe and staining protocol/Colony-Forming Assay. The volume of tracers and reagents used is missing.
We added information to the text.
2.4. Live/Dead Cell Assay. Information on the number of cells counted is missing.
We measured the areas of fluorescent staining. The term “the number of cells” was mistakenly used and now it is corrected.
3.Results.
3.1. Effects of Collagen on Cytotoxicity of FOLFOX.
Figure 2A. The colony-forming assay. According to the assumption of the test, the colonies of surviving cells should be counted and a cell survival curve should be plotted. Only the optical density of crystal violet is given in the figure.
We added the number of colonies in the Figure 2.
Figure 2B. In the discussion of the results, the authors write about the change in the number of live and dead cells. However, no numerical or percentage values were given.
Figure 3. The figure caption lacks information about the cell incubation time during which dyes colocalization was observed.
We added time of incubation to Figure 3 legend.
Figures 4 and 5. The figures show additional incubation times of 1 h and 5 h in addition to 24, 24 and 72 h. What were these times used for? The background in the presented photographs should also be reduced (cut off) (Figure 4A).
The aim of our study was to investigate both the early and late response of cancer cells, that is why we used different time points starting from 1 h.
Round 2
Reviewer 1 Report
Comments and Suggestions for Authors
The authors have addressed my comments
Comments on the Quality of English Language
n/a
Author Response
Thank you for your help
Reviewer 3 Report
Comments and Suggestions for Authors
1. Authors should prepare the manuscript more carefully in terms of graphics.
- chapter numbering should start with the number 1 (according to Microsoft Word Template for IJMS).
-different font sizes were used in the manuscript (Figure 2, line: 288, 389, 553, etc.).
2. According to the previous comment, only significance values should have been added in Figure 2 (MTT viability test). There is no need to show Table S1 with the Post-Hoc results analysis.
3. My earlier comment regarding the MTT test referred to showing the results of cell viability after using drugs according to the FOLFOX regimen. The MTT assay should corelate with the other analyzes shown in the remaining figures in terms of incubation time. Moreover, I do not understand why the authors replaced the previous results from the MTT test performed according to the FOLFOX regimen with the results after oxaliplatin.
4. In response to the review, the authors write that pH and metabolic monitoring experiments were performed at 1 hour, 5 hours, 24, 48, and 72 hours to examine both early and late effects of treatment.
Therefore, I believe that in the manuscript, in order to show the full mechanism of action of drugs in the FOLFOX system, all data should be standardized with the same incubation times. Therefore, it is necessary to complete the experimental data in Figures 2 and 3.
5. Results from the clonogenic assay should be shown as a survival curve, not a colony count (Figure 3C).
6. There is a caption for Figure S1 in the supplementary materials, which is not present.
Summary, the manuscript is interesting, but still requires refinement.
Author Response
- Authors should prepare the manuscript more carefully in terms of graphics.
- chapter numbering should start with the number 1 (according to Microsoft Word Template for IJMS).
-different font sizes were used in the manuscript (Figure 2, line: 288, 389, 553, etc.).
We thank the Reviewer for careful reading of the Manuscript. We used the IJMS template to prepare the revised version.
- According to the previous comment, only significance values should have been added in Figure 2 (MTT viability test). There is no need to show Table S1 with the Post-Hoc results analysis.
We thank the Reviewer for this recommendation. We added significance values in Figure 2 and removed Table S1 from Supplementary.
- My earlier comment regarding the MTT test referred to showing the results of cell viability after using drugs according to the FOLFOX regimen. The MTT assay should correlate with the other analyzes shown in the remaining figures in terms of incubation time. Moreover, I do not understand why the authors replaced the previous results from the MTT test performed according to the FOLFOX regimen with the results after oxaliplatin.
We agree with the Reviewer that correlation of viability assay with other results can improve the understanding of the effects of the FOLFOX regimen. Since MTT-assay is not applicable in the 3D collagen gels model, we included the live/dead cells assay results for all time-points instead. Figure 3 was modified accordingly. The data on correlation analysis are included in Supplementary Material.
In Figure 2 the axis X was renamed according to the FOLFOX combinations, and the Table with the concentrations of the drugs used in a combination was added.
- In response to the review, the authors write that pH and metabolic monitoring experiments were performed at 1 hour, 5 hours, 24, 48, and 72 hours to examine both early and late effects of treatment.
Therefore, I believe that in the manuscript, in order to show the full mechanism of action of drugs in the FOLFOX system, all data should be standardized with the same incubation times. Therefore, it is necessary to complete the experimental data in Figures 2 and 3.
We agree with the Reviewer and completed the data with live/dead cells assay results for all time-points. Actually, the main idea of this research was to investigate the effects of collagen on the efficacy of standard chemotherapy in the FOLFOX regimen, but not the mechanism of action of drugs in the FOLFOX system. So we sincerely believe that the provided full data on cells viability upon treatment will be sufficient.
- Results from the clonogenic assay should be shown as a survival curve, not a colony count (Figure 3C).
Here, we would like to specify that the clonogenic assay was used only as a complementary assay to the live/dead assay to confirm the greater efficacy of FOLFOX in the absence of collagen compared with the model where collagen is present. Therefore, we performed it for only one time-point - 24 h of incubation, when the effects of the treatment are evident. Similarly to live/dead cells assay, the clonogenic assay demonstrated that collagen reduced the cytotoxic effects of FOLFOX.
- There is a caption for Figure S1 in the supplementary materials, which is not present.
The caption was removed.
Round 3
Reviewer 3 Report
Comments and Suggestions for Authors
The manuscript has been corrected on the points I raised. I am recommending it for further processing after minor corrections.
1. Statisticali significances should be entered in Figure 1 E,F.
2. To clarify the time points at which the experiments were performed, I propose including in the manuscript an explanation regarding the clonogenic test, which was sent in response to the review.
3. Figure 6. I suggest placing the caption "drugs combination" of the X axis instead of "combination".
Author Response
- Statisticali significances should be entered in Figure 1 E,F.
We thank the Reviewer for this recommendation. We added significance in Figure 1 E, F
- To clarify the time points at which the experiments were performed, I propose including in the manuscript an explanation regarding the clonogenic test, which was sent in response to the review.
We thank the Reviewer for this recommendation. The following text was added to the main text “We used the colony‐forming assay only as a complementary test to the live/dead assay to confirm the greater efficacy of FOLFOX in the absence of collagen compared with the model where collagen is present. The test was performed once in 24 hours of drugs expo-sure.”
- Figure 6. I suggest placing the caption "drugs combination" of the X axis instead of "combination".
We thank the Reviewer for this recommendation. We changed the caption of X axis in Figure 6 to the “drugs combination”